# Wheat Straw Biochar as a Specific Sorbent of Cobalt in Soil

**DOI:** 10.3390/ma13112462

**Published:** 2020-05-28

**Authors:** Agnieszka Medyńska-Juraszek, Irmina Ćwieląg-Piasecka, Maria Jerzykiewicz, Justyna Trynda

**Affiliations:** 1Institute of Soil Science and Environmental Protection, Wroclaw University of Environmental and Life Sciences, Grunwaldzka 53, 50-357 Wrocław, Poland; irmina.cwielag-piasecka@upwr.edu.pl; 2Faculty of Chemistry, Wroclaw University, Joliot-Curie 14, 50-383 Wrocław, Poland; maria.jerzykiewicz@chem.uni.wroc.pl; 3Department of Experimental Biology, Wroclaw University of Environmental and Life Sciences, Norwida 27b, 50-375 Wrocław, Poland; justyna.trynda@upwr.edu.pl

**Keywords:** biochar, wheat straw, sorbent, cobalt, copper, soil

## Abstract

There is an urgent need to search for new sorbents of pollutants presently delivered to the environment. Recently biochar has received much attention as a low-cost, highly effective heavy metal adsorbent. Biochar has been identified as an efficient material for cobalt (Co) immobilization from waters; however, little is known about the role of Co immobilization in soil. Hence, in this study, a batch experiment and a long-term incubation experiment with biochar application to multi-contaminated soil with distinct properties (sand, loam) were conducted to provide a brief explanation of the potential mechanisms of Co (II) sorption on wheat straw biochar and to describe additional processes that modify material efficiency for metal sorption in soil. The soil treatments with 5% (v/w) wheat straw biochar proved to be efficient in reducing Co mobility and bioavailability. The mechanism of these processes could be related to direct and indirect effects of biochar incorporation into soil. The FT-IR analysis confirmed that hydroxyl and carboxyl groups present on the biochar surface played a dominant role in Co (II) surface complexation. The combined effect of pH, metal complexation capacity, and the presence of Fe and Mn oxides added to wheat straw biochar resulted in an effective reduction of soluble Co (II), showing high efficiency of this material for cobalt sorption in contaminated soils.

## 1. Introduction

In recent decades, industry’s reliance on cobalt as a material essential for enabling technological development has caused considerable growth in the use of cobalt and accidental release of this metal into the environment. Metal ore mining and the smelting process (mainly copper, zinc, lead, and cobalt), alloys and chemicals containing cobalt (Co), sewage effluents, and urban and agricultural runoff (phosphate fertilizers and pesticides) [1] have been described as the main sources of cobalt pollution in the environment. Since 50% of cobalt produced globally is found in rechargeable lithium-ion batteries [2], the electronic devices industry and its reliance on cobalt should be considered to be a new environmental threat. The scale of cobalt release to environmental components is not well recognized. Elevated concentrations of cobalt in soil and groundwater occur locally depending on the local geology or atmospheric deposition from metal ore mining and smelting sites, making the problem insignificant. However, cobalt can be easily transferred by air deposition into soil or leached to groundwater, affecting crop quality and food safety [3]. Cobalt plays a significant role as a constituent of vitamin B12, however, excessive exposure has been shown to induce various adverse health effects [4]. The occurrence of xenobiotic with unknown impacts on the environment and human health brings new challenges to risk reduction of elemental transfer to food chains. Among the variety of methods of soil remediation, the application of chemical amendments to polluted soil, leads to reduced environmental risks of heavy metals through several chemical mechanisms including adsorption [5], precipitation [6], and complexation [7]. According to previous studies, inorganic materials such as lime, zeolite, and phosphate are effective for heavy metals immobilization [8]. There are also organic amendments which can achieve similar efficacy, such as peat, brown coal, and biosolid compost [9,10,11,12]. The application of organic materials is a main strategy for remediation of soil polluted with heavy metals, but this procedure should be considered carefully in the case of Co polluted soils, as raw organic materials can increase mobility of this element in soil due to formation of organic chelates [9]. In a search for the most desired and efficient remediation material for cobalt, biochar should be considered. Due to the presence of a highly-porous structure [13], various functional groups (e.g., carboxyl, hydroxyl, and phenolic groups) [14] on biochar show a great affinity for metal cations [15]. The composition and biochar stability establish the sorption properties of the material. Surface functional groups present on organic carbonaceous phases play the most important role, as they decide about properties of the biochar that are important for heavy metal sorption such as pH, negative charge on the surface, the cation exchange, and surface complexation potential for metals [16]. In addition to organic components, biochars also contain mineral components such as quartz, calcite, sylvite, periclase, and whitlockite [17]. The mineral components of biochars can work as additional sorption sites for metals, ion exchange [18,19], surface complexation [20] and formation of metal precipitates [21] by releasing soluble ions, which include phosphates, sulphates, and carbonates [22,23]. A comparison with other forms of carbonaceous sorbents shows that biochar is a promising adsorbent with lower cost for metal removal from water. Much research has recently been conducted to explore biochar efficiency for heavy metal, including Co removal from an aqueous solution [24,25,26,27]. Most of this research has provided sorption mechanisms for metals as a group, however, a comparison of mechanisms for removal of different metals is necessary to describe biochar capacity for heavy metal sorption. As different metals are present in the environment in different species or valence states under different pH or red-ox conditions, the main mechanisms for their sorption could be different [28]. Cobalt most commonly occurs in the soil as Co (II) and Co (III) ions, however bioavailability and the potential environmental risk of this species in soil is distinct. The behavior of Co, in soils, is influenced to a large degree by the presence of Mn and Fe oxides which are known to have a great affinity for Co, as most of the Co (up to 79%) has been found strongly associated with Fe and Mn oxyhydroxides in soils [9,29]. Co (II) is highly soluble in water, potentially very mobile [30], and bioavailable [4]. Co (III) occurs mainly through surface oxidation of Co (II) on oxyhydroxide minerals [29], which is an important process reducing Co mobility and bioavailability in soils. Many different methods have been dedicated to estimate the efficiency of the material for metal sorption, however, sorbent efficiency could be different in soil as compared with aqueous solutions, as soil properties such texture, organic matter content, pH, or redox conditions have an influence on metal mobility and bioavailability of metal ion, making this matrix more dynamic and interactive. Described soil properties can be modify by sorbent when added to soil. As well as the biochar properties can be changed over time by weathering, leaching, oxidation, or biodegradation processes after remaining in the soil for a period of time. This makes material evaluation for remediation purposes more complex. The present study focuses on wheat straw biochar efficiency for Co sorption in soil. The batch experiment and the long-term incubation experiment with biochar application to contaminated soil provide a brief explanation of the potential mechanisms of Co sorption on wheat straw biochar and describe additional processes that modify material efficiency for metal sorption in soil. 

## 2. Materials and Methods 

### 2.1. Biochar Characteristics

Biochar was produced from wheat straw (WSBC) at the pyrolysis temperature of 550 °C and time remaining in the reactor 60 s. The BET surface area, cation exchange capacity (CEC), pH in deionized water, CNHSO elemental composition, ash and carbonates content (% volume/dry weight), exchangeable cations and anions content (Ca^2+^, Mg^2+^, K^+^, Na^+^, P, NH_4_^+^, NO_3_^−^), and the total contents of trace elements (Co, Mn and Fe) were determined to describe the properties of the material. The total surface area was determined using a BET (Brunauer, Emmett and Teller) specific surface area analyzer Gemini VII 2390 Series (Micrometrics Instruments Corporation, Norcross, GA, USA). The cation exchange capacity and exchangeable cations (Ca^2+^, Mg^2+^, K^+^, Na^+^, and NH_4_^+^) were determined according to the modified method described by Munera-Echeverri et al. [31] and analyzed on a microwave plasma-atomic emission spectrometer MP-AES 4200 (Agilent Technologies, Santa Clara, CA, USA). Exchangeable P was analyzed on the MP-AES 4200 after sample extraction according to Olsen et al. [32,33], as 0.5 M sodium bicarbonate (NaHCO_3_) solution at a pH of 8.5 had a similar pH to WSBC. This extractant decreased calcium in solution (through precipitation of calcium carbonate), and this decrease enhanced the dissolution of Ca-phosphates. The nitrate content was analyzed according to the ISO 14256-1:2003 procedure on a UV-Vis Cary 60 (Agilent Technologies, Santa Clara, CA, USA). The pH values were measured at a ratio of 1:5 (w/v) in deionized water after the sample was shaken for 1 h at 130 rpm with a calibration check pH meter (Mettler Toledo, Columbus, OH, USA). The ash content was determined by weight loss after combustion at 750 °C for 6 h in a muffle furnace according to ASTM D7348-13 [34]. CaCO_3_ was determined following the Scheibler method with a calcimeter [35]. The elemental composition (CHNSO) was analyzed on a CHNS analyzer (CE Instruments, Hindley Green, UK), and the O content was calculated from the difference. The total content and exchangeable forms of metals (Co, Cu, Fe, and Mn) were analyzed on a microwave plasma-atomic emission spectrometer MP-AES 4200 (Agilent Technologies, Santa Clara, CA, USA), respectively, after microwave sample digestion in 70% nitric acid (1:10 w/v ratio) in a digestion microwave system StartD (Milestone Srl.Sorisole, Italy) and extraction with deionized water (1:25 w/v ratio).

### 2.2. Metal Sorption Mechanism Analysis

Previous studies of Co sorption in soil have showed that Co and Cu as divalent cations can compete for sorption sites, especially in excess of Cu^2+^ in multi-contaminated soils, as described by Muyumba et al. [36]. In the metal sorption experiment, both metals were used to simulate natural conditions in contaminated soils and possible interactions of the Co and Cu ions. The FTIR spectroscopy was used to compare potential changes in the functional groups of metal-loaded biochars with the biochar samples before the Co or Cu sorption. The Co and Cu sorption on the wheat straw biochar was determined by the simplified batch equilibrium method according to the OECD 2000/106 protocol [37]. To compare the effect and probable interaction between cobalt and copper that could occur in multi-contaminated soils the following three solution were used in the batch experiment: (1) Co (II) acetate, (2) Cu (II) acetate, and (3) Co (II) + Cu (II) mix of both salts. Briefly, 5 g of each salt was diluted in 500 mL of deionized water and set overnight to reach equilibrium. The pH of each solution was measured before and after the batch experiment to determine if the pH change occurred after biochar BC addition which would affect sorption conditions. One gram of wheat straw biochar and 20 mL of each solution were placed in 50 mL volume polypropylene falcon centrifuge tubes. All samples were prepared in three replicates of each treatment. The sealed samples were placed on the rotary shaker Multi RS-60 (Biosan, Riga, Latvia) at 80 rpm and 22 ± 0.5 °C for 24 h. Sorption equilibrium was reached within less than 24 h. Then, samples were centrifuged for 25 min at 10,000 rpm to separate the biochar from the solution according to procedure described by Ćwieląg-Piasecka et al. [38]. The biochar samples were washed three times with 20 mL of deionized water and prefiltrated on Munktell No. 2 filter papers (Ahlstrom Munksjö, Helsinki, Finland) to rinse off excess metal cations. The biochar samples were dried in an oven drier at 60 °C for 6 h to prepare pellets for Fourier transform infrared spectra (FT-IR). The FT-IR analysis of the wheat straw biochar samples were recorded using a Vertex 70 FT-IR spectrometer (Bruker, Billerica, MA, USA) on KBr pellets (about 1 mg sample for 400 mg of KBr) according to the standard method used for sample preparation for FT-IR spectra analysis. The incubation experiment with multi-contaminated soils was a pot experiment with 24, four-liter pots (approximately 3 kg of soil each). Two soil types, sand and loam, were set as 12 control pots, six for each soil type. A similar 12 pots were amended with a dose of BC (5.0% w/v), six for each soil type. The soil mixtures were incubated for two years, keeping the humidity of the pots at 60% of maximum water holding capacity. After the time period, the soil samples were collected from each pot, air dried, and sieved (<2 mm), and sequential extraction of Co was performed. The existence of possible precipitates of Co after metal sorption was checked using a scanning electron microscope (SEM) (Bruker, Billerica, MA, USA) coupled with an energy dispersive X-ray analyzer (EDX) (Bruker, Billerica, MA, USA). The biochar particles for SEM-EDX (scanning electron microscopy with energy dispersive X-ray spectroscopy) analysis were separated from the soil by progressive sieving of the soil-biochar mixture. Biochar fraction <1 mm was used to determine elemental composition of the material surface and the distribution of ions [39]. For the morphological observations, SEM was applied according to Michalak et al. [40].

### 2.3. Soils Sampling

Ten soil samples were collected from the topsoil horizon (0–20 cm) at afforested sites at different distances and locations from the copper smelter in SW Poland (16°01′40”N, 51°45′09”E), expecting cobalt enrichment from smelter emissions. In the samples, pre-scanning studies for a wide range of heavy metals, including cobalt, were done on a microwave plasma-atomic emission spectrometer MP-AES 4200 (Agilent Technologies, Santa Clara, CA, USA) after microwave sample digestion in 70% nitric acid (1:10 w/v ratio). The total concentration of cobalt remained in a wide range from 4.5 to 74 mg/kg, depending on the distance from the smelter and the soil type. Two soils with the highest Co concentrations, differing in texture, were chosen for the incubation experiment. Large soil samples, 15 kg each, were collected of Cutanic Luvisol and Fulvic Brunic Arenosol according to the FAO (Food and Agriculture Organization of the United Nations) guidelines [41] at a distance of approximately 2 km from the potential emission source. The copper smelter has been in operation since 1968, becoming one of the most important sources of airborne heavy metal pollution in this area. Polymetallic deposits contain about 1.4% of Cu, however significant amounts of Co-barring minerals can be also found in the ore, mainly Cu-Co, Ni-Co, and As-Co minerals. As the ore is excavated, floated, and smelted, these processes become significant sources of airborne cobalt emission. Unfortunately, no data on cobalt and other rare metals are available for this area. The basic soil properties were examined using common methods described by [42]. To simplify the descriptions further, soil samples were named L for Cutanic Luvisol and S for Fulvic Brunic Arenosol, as well as C for the control, and WSBC for wheat straw biochar. The initial physicochemical properties and total content of cobalt of the studied soils are presented in Table 1. The Cutanic Luviosol Control (LC) soil sample was loamy and neutral in nature, non-saline, with average European soil total carbon content and high CEC. The Fulvic Brunic Arenosol Control (SC) soil sample was sandy and acidic, non-saline, with typical sandy soil low carbon content and low CEC. The total content of Co in the LC soil sample was higher than that in the SC soil sample, respectively, 67 and 26 mg/kg. In both soil samples, the total Co concentrations were higher than that for European soils, i.e., <9.3 mg/kg [8,38,39]. Naturally, higher Co contents were observed in soils around ore deposits, as the distance from the smelter to mining sites is about 40 km, we indicated this element as an airborne pollution from the copper smelter.

### 2.4. Cobalt Immobilization Analysis in Soil

After soils were incubation with 5% (w/v) wheat straw biochar, the (Community Bureau of Reference (BCR) sequential extraction procedure was applied to measure the following four fractions of cobalt in the tested soil: exchangeable and bound to carbonates (Fraction 1), reducible or bound to Fe and Mn-oxides (Fraction 2), oxidizable or bound to organic substances (Fraction 3), and residual (Fraction 4). Acetic acid, hydroxyl ammonium chloride, hydrogen peroxide plus ammonium acetate, and aqua regia stages of the sequential extraction procedure were applied to the soil samples, respectively [43,44,45]. The contents of cobalt, extracted during the BCR procedure, were measured on a microwave plasma-atomic emission spectrometer MP-AES 4200 (Agilent Technologies, Santa Clara, CA, USA). Data are provided as an average result from triplicate with the relative standard deviation (RSD), calculated by MP Expert Software Agilent Technologies. The maximum relative standard deviation (RSD) between replicates was set to 5%. Values that were above 5% were not included in the statistical analyses. To avoid analytical errors, standard solutions (from LGC Standards Ltd., UK) for MP-AES 4200 were used for calibration and certified reference materials as follows: RTH 953 Heavy Clay Soil from LGC Promochem (LGC Standards Ltd., Teddington, UK), total Co content 14.7 mg/kg and CRM055 (Honeywell Fluka, Charlotte, NC, USA) with Co content of 97 mg/kg were analyzed with every sample set. The recovery of Co from Certified Reference Material (CRM) was 89–94% and the maximum values of RSD were 2.6%. Detection limits were 0.01 mg/kg of Co in the soil samples. 

### 2.5. Statistical Analysis

The metal sorption batch experiments were performed in triplicate and the soil sorption experiments were performed in six replicates. The data are presented as the mean values with the relative standard deviation (RSD). Student’s *t*-tests were used to test for significant differences in cobalt fractions between biochar treated and untreated soils (*p* < 0.05). The obtained data were compiled using Microsoft Excel 2016 and Statistica Statsoft 13.3.

## 3. Results

### 3.1. Sorbent Characteristics

The elemental composition of the tested wheat straw biochar is presented in Table 2. The results indicated that the C content in the WSBC was 63.3%, followed by O content of 33.46%, N content of 0.74%, H content of 2.2%, and very low S content of 0.037%. The calculated H/C and O/C molar ratios are the indicators of BC aromaticity and polarity, respectively. It is assumed that BCs produced at a temperature higher than 400 °C should be characterized by an H/C ratio lower than 0.5 and decrease with the raising pyrolysis temperature below 0.3, which is an indicator of highly aromatic ring systems. In the case of the investigated BC, produced at 550 °C, the H/C ratio falls within the 0.3–0.5 range which could indicate a decreased fraction of original wheat residues. The obtained BC molar ratios (Table 2) emphasize the presence of aromatic structural features and reduced content of O-containing polar functional groups on BC surface (low molar O/C ratio and polarity index) [38]. 

Sorption properties of tested wheat straw biochar are given in Table 3. The average specific surface area (SSA) of the tested WSBC was 256 g/m^2^, with the cation exchange capacity (CEC) of 63 cmol_c_/kg. The total ash content was 32.4% and the contribution of calcium carbonates was 3.07% (w/dw), however almost 90% of exchangeable cations in CEC was a K^+^, followed by Ca^2+^, Mg^2+^, and a very low content of Na^+^ and NH_4_^+^ (Table 3). The content of exchangeable phosphorus, mainly in forms represented by Ca-phosphates (Olsen P) was 265 mg/kg, which was less than 9% of the total phosphorus in tested wheat straw biochar. The content of NH_4_^+^ and NO_3_^−^ was very low, and less than 0.1% of total nitrogen was in exchangeable forms after sample extraction with 1 M KCl. The total content of sulfur was 0.037–0.042% (Table 1), suggesting that the contribution of sulfate forms in WSBC was negligibly low. The low content of cation and anion in exchangeable forms after extraction with weak extractants could be attributed to the very high pH (9.86) of biochar. Similar to multi-contaminated matrixes such as the soils from the copper smelter area, competition, between Co (II) and other metal cations, can occur and some of the metals present on the biochar surface can be exchanged, and therefore the potential contribution in this process of Cu, Fe, and Mn was analyzed. The results showed that only 2.6% of Cu, 2.8% of Fe, and 4.1% of Mn in the tested biochar were in readily exchangeable forms. The total content of Co and exchangeable Co forms showed very low potential contributions of biochar-derived Co, in the sorption/desorption processes (Table 4). However, high content of exchangeable Cu^2+^ in the tested soil could induce competition between both divalent cations for sorption sites on the biochar surface, which was shown in the batch experiment.

### 3.2. Metal Ions Sorption on Biochar

The FT-IR analysis showed differences between pure and spiked with Co (II), Cu (II), and mix Co (II) + Cu (II) biochars (Figure 1). From the presented spectra, the most probable mechanism of metal ions binding can be related to the oxygen containing groups on wheat straw biochar surface, as the most characteristic changes occurred at vibrations 3428, 1624, and 1420 cm^−1^. The metal ions in metal spiked biochars decreased the intensity of the peaks at 3428 cm^−1^ stretching vibrations of the OH (H-bonding) groups. This change confirms that the O-H groups take part in Co (II) and Cu (II) complexation on wheat straw biochar surface. The carboxyl peak observed for a pure wheat straw biochar at 1624 cm^−1^ was shifted to much smaller values, i.e., 1583 cm^−1^ or 1570 cm^−1^ in the spectra of BC treated with salts (Figure 1).

The decrease in the wavenumber of the peak 1624 cm^−1^, characteristic for C=O carboxylic group, can be explained by the interaction with Co (II) and Cu (II) ions with free carboxyl groups on the biochar surface and change to carboxylates, which indicates the important role of carboxyl groups on the biochar surface in metal binding. To provide more details about the type of the metal binding to carboxylic group on the biochar surface at 1624 and 1420 cm^−1^, a calculation of Δ according to Nakamoto [46] was performed (Table 5).

According to the calculations, carboxylate could coordinate metal ions on three different modes, i.e., unidentate, chelating (bidentate), and bridging [47]. The calculated values of Δ indicate creation of bridging complexes, where two metal ions are involved in the binding of one carboxylic group. Some of the metal coordination sites can be associated with H_2_O or OH groups. The results of the FT-IR analysis also showed that metal ions complexation could be related to an abundance of carbonates and polysaccharides such as moieties in biochar, since changes of peak at 1080 cm^−1^ related to Si–O, C–O, and S=O groups were observed in metal spiked biochar. Very low concentrations of sulfur in biochar (Table 2) can limit metal ion coordination with S=O groups on the BC surface, however, the Si–O groups can be involved in the process. The very strong peak, 464 cm^−1^, can be attributed to vibrations of many moieties. This band could appear when H_2_O is one of the ligand in the complex or Cu–O–H deformations [46].

### 3.3. Cobalt Immobilization in Soils

Wheat straw biochar was applied to soils to verify the hypothesis that application of this material can immobilized cobalt ions in soil. Copper fractionation for tested soils was described in the previous paper [48] and is not be discussed in these results. Figure 2 shows the contribution of Co forms in the control sand and the loam and soils treated with 5% (v/dw) wheat straw biochar. Regardless of the type of soil, the Co contribution of the individual fractions was similar as follows: F1 < F2 < F3 < F4; however, significant differences (p < 0.05) were observed in fraction content between soils. In both tested soils, mobility of Co was high (more than 10%) as compared with other analyzed metals in tested soils [48]. 

The application of 5% (w/v) wheat straw biochar affected cobalt speciation in both tested soils. In sand + 5% WSBC treatment, a significant decrease in exchangeable cobalt Fraction F1 (F1) from 17.4% to 7.3% was determined. The observed reduction in Co content in F1 was mainly balanced by their increased content in Fraction 2 (F2), reducible or bound to Fe and Mn-oxides (Figure 2) by 9.4%. Some of the cobalt forms were also shifted from Fraction 3 (F3), oxidiziable or bound to organic substances and Fraction 4 (F4), becoming residual fraction non-bioavailable or not prone to leaching [49]. In the loam + 5% WSBC treatment, the biochar application did not decrease Co speciation in Fraction 1, however significant amounts of Co were shifted from Fractions 2 and 3 to Fraction 4, increasing cobalt in residual fraction by 40%.

### 3.4. SEM-EDX Analysis of Biochar

The application of SEM-EDX proved that the metal ions can be bound on the biochar surface. Examination of the SEM-EDX element map, Figure 3, presents the relative increase in surface concentration of Co on biochar particle (<1 mm), shown by the image brightness after biochar incubation in contaminated soil. The location of Co “hot spots” on the biochar surface corresponded to sites with increased surface concentrations of Fe and Mn and oxygen functional groups.

## 4. Discussion

Biochar application can have a direct or indirect impact on cobalt immobilization/mobilization process in soil. Indirectly, biochar can affect soil sorption properties and pH, reducing the presence of metal in exchangeable and soluble forms in soil solution. The direct effect can be related to biochar properties such as sorption capacity, oxygen functional groups, and mineral components content (carbonates, phosphates, Mn and Fe oxides) increasing or supporting soil sorption capacity for cobalt. All mentioned processes were considered in our study. The tested wheat straw biochar had high SSA as compared with other straw-derived biochars produced under similar conditions, as porosity of biochar significantly increases between 400–600 °C [50]. Gul et al. [51] characterized wheat straw biochar with SSA from 178 to 184 m^2^/g, depending on pyrolysis (slow or fast), although, in most research, lower values can be found with good efficiencies for heavy metal removal [52,53]. Cation exchange capacity was low, as this property usually decreases with the higher temperatures of pyrolysis [1]. Wheat straw biochar had a very high pH, most likely due to the much higher potassium content found in the straw biochars as compared with wood derived materials, which was also described in another study [54]. The wheat straw biochar had a high ash and carbonates content, although very low content of nitrates, sulphates, and phosphates, which has also been indicated by other authors studying straw-derived biochars [51,54]. Biochar properties affect soil properties when applied to soil [50], modifying conditions of heavy metal mobility (mainly CEC and pH). Wheat straw biochar failed to change the soil CEC significantly (p > 0.05) in both tested soils. Normally, biochars are considered to develop more oxygen-containing functional groups, and hence increase CEC and negative charge of soils [13,19,55]. Our study suggested that an increase of soil functional groups after WSBC was not enough to change sorption properties of soil, which was in agreement with a study by Qi et al. [56]. Wheat straw biochar changes soil pH significantly (p > 0.05), but only in acidic sandy soil, which could affect cobalt mobility indirectly. Soil pH is one of the most important environmental factors affecting sorption of toxic metal [57]. In other experiments with wheat straw biochar, cobalt sorption from aqueous solution was described as pH dependent, i.e., very low at pH values from 2.0 to 4.0 and high between pH values of 5.5 and 8.0 [43,58], similar to the pH of tested soil samples after WSBC application. A lower pH, such as in sandy soil, enhanced cobalt mobility and potential bioavailability for plants, which was also observed in other studies on acidic soils [59,60]. As our study did not show a significant impact on soil sorption capacity, we focused on biochar characterization as some direct mechanisms such as surface complexation or supported sorption by addition mineral compounds like Fe and Mn oxides, could be related to decreased mobility of cobalt in the tested soils. The results of the FTIR analysis showed that the tested wheat biochar had the capacity for Co complexation with oxygen-containing surface functional groups, mainly carboxylic C=O and hydroxyl groups H–O, similar to other divalent cations. Depending on the pH conditions, cobalt typically to other divalent cations, can be hydrated in a soil solution sharing similar sorption mechanisms, i.e., cation exchange, surface complexation, and precipitation [21]. Similar results for active Co chemisorption by hydroxyl functional groups were described by Liu et al. [61]. Sun et al. [27] showed that oxygen-containing surface functional groups, for example, C–O, C–O–C, and C=O increased biochar capability for element immobilization [62], suggesting that these groups play important roles in metal sorption. The quantities and quality of functional groups on biochar surface vary, depending on biochar production conditions and feedstock types used. However, biochars produced at higher temperatures (>500 °C) have higher surface area and porosity, but lower abundance of functional groups, primarily due to the higher degree of carbonization [28]. Zhang et al. [63] observed decreased contribution of O-alkyl carbon from 20% to 54% to 7% to 13% for wheat straw biochars, as temperature increased from 200 to 600 °C, and at >300 °C aromatic structures are dominant [64]. These findings are in agreement with the CHNSO analysis and molar ratios obtained during raw material analysis in this study, emphasizing the presence of aromatic structural features and reduced content of O-containing polar functional groups on WSBC surface. 

As oxygen-containing functional groups are predominant mechanisms of divalent cations the result of our study suggests that biochars produced at lower temperatures or oxidized during pretreatments [65] have better efficiency for Co^2+^ removal as compared with materials produced at high temperatures with a more aromatic structure. However, biochar applied to soil vs. solution undergoes many abiotic and biotic processes causing sorbent oxidation, called biochar aging [66], which can result in increased sorption capacity for cations after time remaining in soil. Aging in soil leads to surface changes [67] and generation of new functional groups on its surface [68]. Uchimiya et al. [67] described that in the presence of soil, the importance of oxygen-containing groups on biochar surfaces in cation sorption strongly depended upon the inherent sorption capacity of soil. Wang et al. [66] observed that decarboxylation of surface functional groups on biochar surface, when added to soil, increased soil pH, but these groups could also affect complex metal cations in the soil solution and reduce bioavilalbility, which is in agreement with our study. 

Findings of our study showed that wheat straw biochar can support soil sorption complex acting as a source of several mineral components, for example, Fe and Mn oxides [69,70], silica, carbonates, and phosphates [71], increasing cobalt immobilization/precipitation in soil. The Fe and Mn oxides were not specifically analyzed by XRF in the present study, however, interaction between the Fe and Mn ions and Co was observed after biochar incubation in soils by SEM-EDS analysis and sequential extraction of Co from tested soils where cobalt fraction bound to Fe and Mn oxides increased significantly (*p* > 0.05) in biochar treated soils. The Mn and Fe oxides and organometallic moieties such as Fe–O–C can be formed on the biochar surface during pyrolysis. Most of the Fe in biochar is present in crystalline phases ranging from zerovalent iron to ferric oxides [72]. The role of hydrous oxides of iron, manganese, and clay minerals as Co sorbent has been described recently [73,74,75,76,77]. As these reactions are strongly pH dependent, under alkaline conditions in loam +5% WSBC treatment, cobalt could coprecipitate as Fe and Mn secondary oxides, shifting Co to residual forms, which has also been observed in other studies [43]. Kabata-Pendias [9] described that cobalt precipitation on Mn oxide surface increased under alkaline conditions forming very stable hydroxyl species Co(OH)_2_ which could explain the high stability of Co forms after WSBC application to loam soil. Pan et al. [24] described that Co immobilization on biochar could be dependent on mineral composition and the content of carbonates, phosphates, and calcium hydroxyapatite (CaHA), and suggested that cobalt could be rapidly exchanged with calcium decreasing element in the solution. This mechanism was also visible in the FT-IR analysis, however, analysis exchangeable cation extraction from tested soil did not indicated any significant changes. Figure 4 presents the described conceptual model and possible mechanisms of cobalt sorption on wheat straw biochar.

The results of our study showed that wheat straw biochar has good removal efficiencies in single-metal systems determined in a batch experiment with cobalt salts in the solution. However, the capacities of biochar for cobalt sorption can be modified in multiple-metal systems due to the competition between the heavy metals present in soil and lower stability of the material due to cation exchange and the surface oxidation process that biochar undergoes under soil conditions. As biochar can affect soil properties, changing conditions of metal immobilization, and as soil conditions can have an impact on surface properties of biochar, predictions about biochar efficiency for metal sorption in soil are difficult and need further recognition. 

## 5. Conclusions

The results of the experiments showed that wheat straw biochar is an efficient sorbent of Co^2+^, decreasing mobility and availability of this element in soil.The dominant mechanisms of cobalt sorption on biochar surface are related to oxygen functional groups present on the biochar surface, mainly carboxylic and hydroxyl groups.Biochar can support soil sorption complex with mineral components such as Fe and Mn oxides which increase efficiency of cobalt immobilization.The efficiency of cobalt removal from soil can be modified by biochar oxidation and interactions with soil constituents which makes the process more complex as compared with cobalt removal from aqueous solutions.The results from this study suggest that application of biochar is a feasible strategy for remediation of cobalt contaminated soils, reducing health risks related to human exposure to Co from anthropogenic sources.Further and more complex studies are necessary to recognize the problems of cobalt soil pollution, potential risks, and remediation solutions, including biochar application.

## Figures and Tables

**Figure 1 materials-13-02462-f001:**
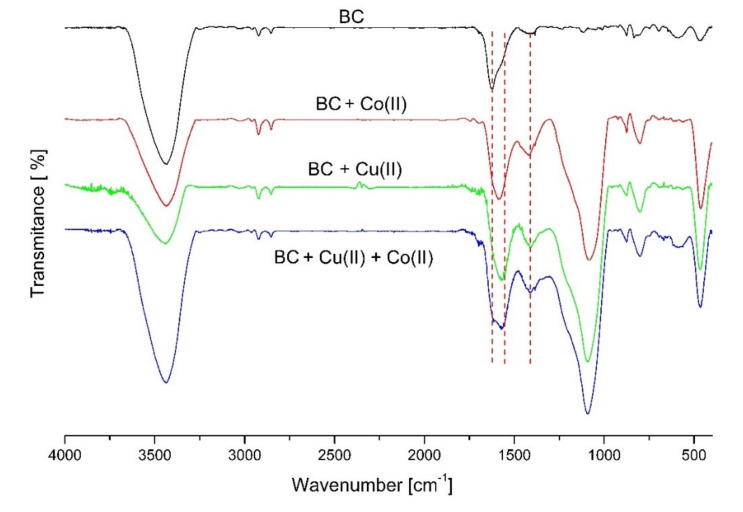
FT-IR spectra of investigated pure wheat straw biochar, biochar spiked with Co (II) salt, biochar spiked with Cu (II) salt, and biochar spiked with a mix of Co (II) and Cu (II) salts.

**Figure 2 materials-13-02462-f002:**
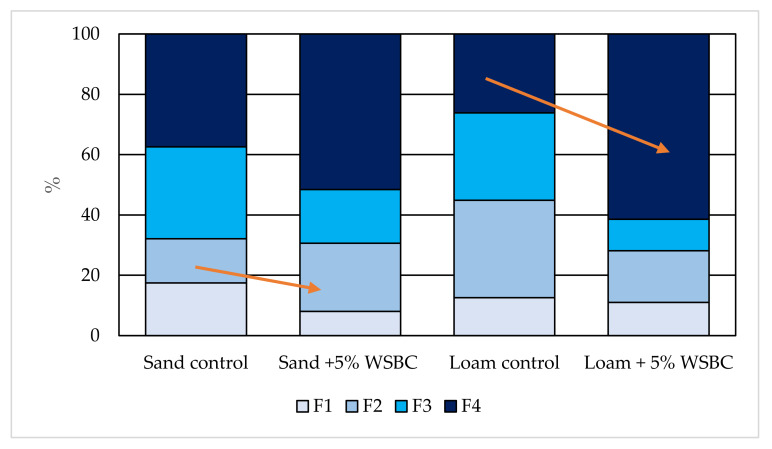
Cobalt speciation in treated and untreated soils. Cobalt fractions: F1, exchangeable and bound to carbonates; F2, reducible or bound to Fe and Mn-oxides; F3, oxidiziable or bound to organic substances; and F4, residual.

**Figure 3 materials-13-02462-f003:**
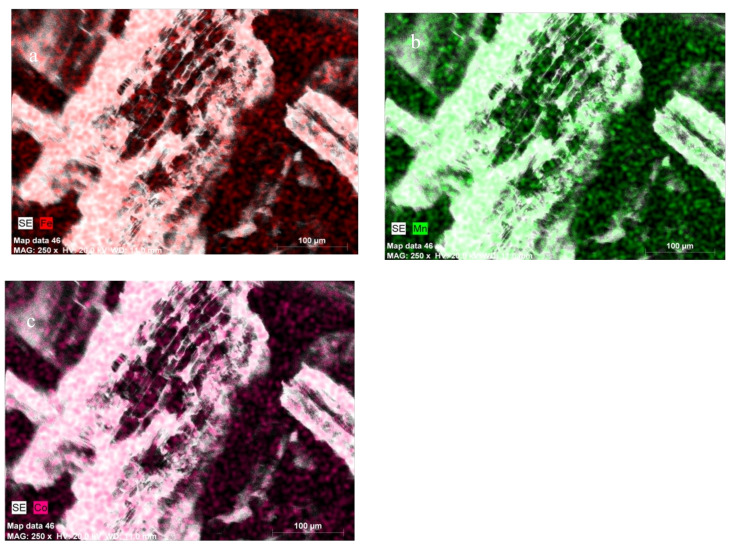
Scanning electron microscope (SEM) images of wheat straw biochar surface and energy dispersive X-ray spectroscopy (EDX) mapping. (**a**) Fe; (**b**) Mn; and (**c**) Co ions, distribution on the biochar produced from wheat straw (WSBC) surface.

**Figure 4 materials-13-02462-f004:**
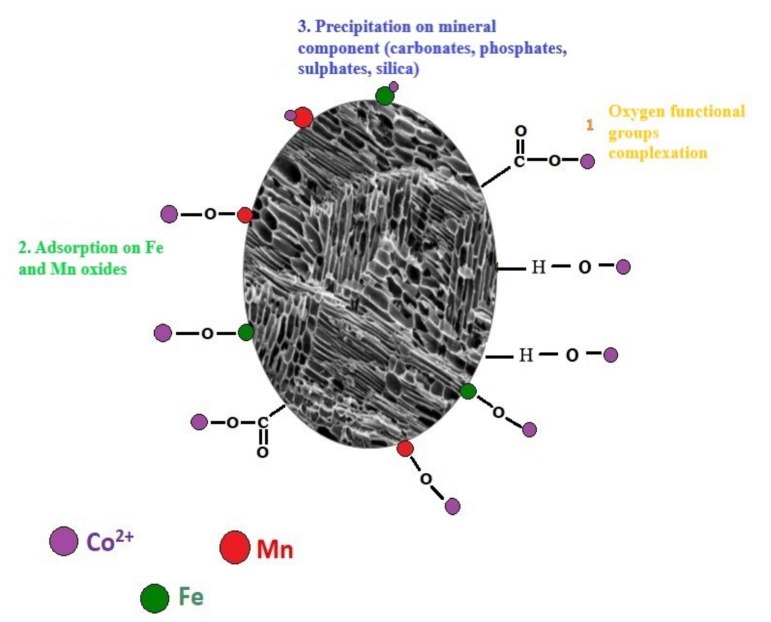
Conceptual model of Co adsorption mechanism on wheat straw biochar surface.

**Table 1 materials-13-02462-t001:** Selected properties of soils from the incubation experiment.

Sample	Depth	Textural Group	Clay %	C_org_ %	pH_H2O_	CEC cmol_c_/kg	Co_tot_ mg/kg
LC	0–20 cm	Silt loam	4	1.12^1^ ± 3.5^2^_a_	6.9 ± 11.0_a_	58 ± 8.3_a_	67.0 ± 2.3_a_
SC	0–20 cm	Loamy sand	<1	0.98 ± 4.6_a_	3.9 ± 20.1_a_	5.5 ± 14_a_	26.0 ± 1.8_a_
L + 5% WSBC	0–20 cm	-	-	1.36 ± 3.8_b_	7.3 ± 14.6_a_	63 ± 9.2_a_	67.2 ± 1.7_a_
S + 5% WSBC	0–20 cm	-	-	1.17 ± 2.7_b_	4.6 ± 12.5_b_	6.3 ± 5.8_a_	26.2 ± 1.9_a_

^1^ Mean values (n = 6) and ^2^ RSD values in % (n = 6). Different lowercase letters (a and b) indicate significant differences between WSBC treated and untreated soil within each soil type (p < 0.05).

**Table 2 materials-13-02462-t002:** Elemental composition of pure and spiked with metal salts wheat straw biochar.

Sample	C	N	H	S	O	H/C	O/C	C/N
	% (w/dw)					molar		
WSBC	63.6^1^ ± 1.6^2^	0.74 ± 3.5	2.2 ± 5.2	0.037 ± 8.8	33.46 ± 0.8	0.34	0.52	85.9
WSBC + Cu (II)	63.8 ± 3.4	0.62 ± 3.2	1.9 ± 5.2	0.041 ± 7.4	33.05 ± 1.9	0.29	0.51	102.9
WSBC + Co (II)	64.5 ± 3.4	0.64 ± 3.2	2.13 ± 1.8	0.041 ± 6.8	33.45 ± 1.2	0.33	0.51	100.7
WSBC + Cu (II) + Co (II)	63.9 ± 0.7	0.65 ± 3.7	2.02 ± 5.8	0.042 ± 6.9	33.32 + 1.4	0.31	0.52	98.3

^1^ mean values (n = 3) and ^2^ RSD values (n = 3).

**Table 3 materials-13-02462-t003:** Properties of wheat straw biochar.

pH_H2O_	SSA	Ash_tot_	CaCO_3_	CEC	Ca^2+^	Mg^2+^	K^+^	Na^+^	NH^4+^	NO_3_^−^	P_ex_ *
	m^2^/g	%	%	cmol_c_/kg					mg/L		
9.86 ^1^ ± 6.9 ^2^	256 ± 3.3	32.4 ± 13.3	3.07 ± 19.1	63 ± 22.4	8.38 ± 14.1	5.22 ± 19.6	58.1 ± 26.7	0.76 ± 25.3	0.025 ± 15.1	4 ± 25.1	265 ± 30.1

^1^ Mean value (n = 6) and ^2^ RSD value (n = 6), * Olsen P.

**Table 4 materials-13-02462-t004:** Total and exchangeable forms of metals in wheat straw biochar.

Co_tot_	Cu_tot_	Fe_tot_	Mn_tot_	Co_ex_	Cu_ex_	Fe_ex_	Mn_ex_
mg/kg							
1.1 ^1^ ± 3.2 ^2^	15.3 ± 4.8	1156 ± 1.2	260 ± 3.5	0.03 ± 4.3	0.4 ± 2.7	32 ± 1.8	10.8 ± 3.7

^1^ Mean value (n = 6) and ^2^ RSD value (n = 6).

**Table 5 materials-13-02462-t005:** Calculated values of Δυ (COO^−^).

Sample	ν_a_(COO^−^) (cm^−1^)	ν_s_(COO^−^) (cm^−1^)	Δ = ν_a_(COO^−^) − ν_s_(COO^−^)
WSBC	1624	1420	204
WSBC + Cu (II)	1583	1419	164
WSBC + Co (II)	1570	1409	161
WSBC + Cu (II) + Co (II)	1574	1411	163

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
