# Peer review of "Wheat Straw Biochar as a Specific Sorbent of Cobalt in Soil"

_materials, 2020, doi:10.3390/ma13112462_

Round 1

Reviewer 1 Report

The scope of this manuscript focuses on wheat straw biochar efficiency
80 for Co sorption in soil.

Major comments

  1. The abstract is very general, and there is a lot of irrelevant information. The abstract should be explained and showed the important aspects of work. So, this abstract in the present form is unacceptable.
  2. As the data analysis of this article, more statistical analysis (CV, mean)  should been added, for example, the coefficient of variation of the experimental data should been added and discussed. This lack of statistics is not acceptable in research publications and needs to be completed.

  3. Is the Fisher test appropriate in this study? Argue the choice of this type of test
  4. As the results and discussion in this article, it is better to compare the research result with other similar research result.
  5. Generally the conclusions should be modified, they are too general, in turn. The achievements could be given in points.

Reviewer 2 Report

This is an interesting research paper, in an important field, searching for low-cost alternatives (wheat straw biochar) to remove contaminants from soil, specifically cobalt. Overall, the research methodology is well implemented and the found issues are related more to the paper itself. For instance, the units inconsistency. Additionally, please be careful with units formatting such as spacing between number and unit and the overall format (e.g. in line 137 “mg kg-1”). There are several issues like this throughout the manuscript.

The manuscript needs English language revision. For instance, the content of the abstract is good, but there are several issues with the English language.

The results from EDX analysis presented in Figure show some odd patterns considering the three maps. Please comment.

Reviewer 3 Report

Overall, the research is interesting as the authors have mentioned that application of biochar for only cobalt immobilization is somewhat limited. However, there are few issues for which the manuscript may not be suitable to accept its current version. The points mentioned below should be properly addressed in the revision in order to get this manuscript accepted.

1) The research and the discussion are only based on wheat straw biochar. Authors did not expand the discussion with respect to biochar from other sources. Especially, when biochar feedstock is one of the key factors that control the quantities of functional groups on the surface of biochar. They also did not discuss much about any differences between wheat straw based biochar with other biochars. Thus, in a sense the work is not complete enough. Authors should include a detailed discussion on how this research knowledge can be used with other biochars.

2) The research is also designed to consider only Co(II). Authors did not mention if the knowledge generated in this research could also be used to adsorb Co(III). Although Co(II) is the primary contaminant found in environment, in some cases (e.g. industrial environment) Co(III) contaminations are common. It would be better if authors discuss the potential use of wheat straw biochar to remediate Co(III) contamination.

3) Authors used a few citations (such as e.g. 17) that were not properly explained or justified with respect to the described research. However, they missed to provide citations for a few key information such as “. As 50% of cobalt produced globally is found in rechargeable lithium – ion batteries, electronic devices industry and its reliance on cobalt should be considered as new environmental threaten”.

Reviewer 4 Report

The study presented here is interesting but rather incomplete.

In fact, the authors checked the competition between Co and Cu, but there is no indication on the behaviour of the material in presence of real or near-real samples, which I suppose are also containing common cations such as K, Na, Ca, Mg, and common anions such as Cl, SO4, HCO3/CO3, NO3. These may affect the performances of the system, possibly requiring, for example, a slightly higher operating temperature, or a bit longer contact time, etc... The authors neglected this part but I think it should be added in the manuscript for a complete study, thus the authors should perform at least one experiment in presence of these species to ascertain the quality of the sorbent.
Moreover, it is not clear along the manuscript the contribution of Fe and Mn. They looks similarly uptaken by the biochar, but a comparative study on cross-competition was not performed. It seems than the study is not mostly dedicated to Co, so a more detailed analysis on all the three cations is required, also considering the above-mentioned catiosn and anions.

The authors cited BET mesaurements, but they are absent from the manuscript. The authors should show these analyses as they are important to define the possible porosity features of these materials.

Other formal remarks:

Please check the grammar and remember that only the angle degree (°) can be attached to numbers, all other units of measure must be separated (unless specific journal requirements).

Fig. 1 should highlight the most representative bands and variations, as in the current state it is not very informative.

The F1 to F4 fractions in figure 2 should also be explained in the main text along with the captions.

Round 2

Reviewer 1 Report

The authors have improved the paper according to my comments.

Reviewer 3 Report

The manuscript in its current form may be accepted for publication.

Reviewer 4 Report

The authors have fulfuilled the queries and I am happy with the current version.